# Adaptive Multi-Column Deep Neural Networks with Application to Robust Image Denoising

**Forest Agostinelli**      **Michael R. Anderson**      **Honglak Lee**
Division of Computer Science and Engineering
University of Michigan
Ann Arbor, MI 48109, USA
{agostifo,mrander,honglak}@umich.edu

## Abstract

Stacked sparse denoising autoencoders (SSDAs) have recently been shown to be successful at removing noise from corrupted images. However, like most denoising techniques, the SSDA is not robust to variation in noise types beyond what it has seen during training. To address this limitation, we present the *adaptive multi-column stacked sparse denoising autoencoder* (AMC-SSDA), a novel technique of combining multiple SSDAs by (1) computing optimal column weights via solving a nonlinear optimization program and (2) training a separate network to predict the optimal weights. We eliminate the need to determine the type of noise, let alone its statistics, at test time and even show that the system can be robust to noise not seen in the training set. We show that state-of-the-art denoising performance can be achieved with a single system on a variety of different noise types. Additionally, we demonstrate the efficacy of AMC-SSDA as a pre-processing (denoising) algorithm by achieving strong classification performance on corrupted MNIST digits.

## 1   Introduction

Digital images are often corrupted with noise during acquisition and transmission, degrading performance in later tasks such as: image recognition and medical diagnosis. Many denoising algorithms have been proposed to improve the accuracy of these tasks when corrupted images must be used. However, most of these methods are carefully designed only for a certain type of noise or require assumptions about the statistical properties of the corrupting noise.

For instance, the Wiener filter [30] is an optimal linear filter in the sense of minimum mean-square error and performs very well at removing speckle and Gaussian noise, but the input signal and noise are assumed to be wide-sense stationary processes, and known autocorrelation functions of the input are required [7]. Median filtering outperforms linear filtering for suppressing noise in images with edges and gives good output for salt & pepper noise [2], but it is not as effective for the removal of additive Gaussian noise [1]. Periodic noise such as scan-line noise is difficult to eliminate using spatial filtering but is relatively easy to remove using Fourier domain band-stop filters once the period of the noise is known [6].

Much of this research has taken place in the field of medical imaging, most recently because of a drive to reduce patient radiation exposure. As radiation dose is decreased, noise levels in medical images increases [12, 16], so noise reduction techniques have been key to maintaining image quality while improving patient safety [27]. In this application, assumptions must also be made or statistical properties must also be determined for these techniques to perform well [26].

Recently, various types of neural networks have been evaluated for their denoising efficacy. Xie et al. [31] had success at removing noise from corrupted images with the stacked sparse denoising

autoencoder (SSDA). The SSDA is trained on images corrupted with a particular noise type, so it too has a dependence on a priori knowledge about the general nature of the noise.

In this paper, we present the adaptive multi-column sparse stacked denoising autoencoder (AMC-SSDA), a method to improve the SSDA's robustness to various noise types. In the AMC-SSDA, columns of single-noise SSDAs are run in parallel and their outputs are linearly combined to produce the final denoised image. Taking advantage of the sparse autoencoder's capability for learning features, the features encoded by the hidden layers of each SSDA are supplied to an additional network to determine the optimal weighting for each column in the final linear combination.

We demonstrate that a *single AMC-SSDA network* provides better denoising results for both noise types present in the training set and for noise types not seen by the denoiser during training. A given instance of noise corruption might have features in common with one or more of the training set noise types, allowing the best combination of denoisers to be chosen based on that image's specific noise characteristics. With our method, we eliminate the need to determine the type of noise, let alone its statistics, at test time. Additionally, we demonstrate the efficacy of AMC-SSDA as a preprocessing (denoising) algorithm by achieving strong classification performance on corrupted MNIST digits.

## 2   Related work

Numerous approaches have been proposed for image denoising using signal processing techniques (e.g., see [23, 8] for a survey). Some methods transfer the image signal to an alternative domain where noise can be easily separated from the signal [25, 21]. Portilla et al. [25] proposed a wavelet-based Bayes Least Squares with a Gaussian Scale-Mixture (BLS-GSM) method. More recent approaches exploit the "non-local" statistics of images: different patches in the same image are often similar in appearance, and thus they can be used together in denoising [11, 22, 8]. This class of algorithms—BM3D [11] in particular—represents the current state-of-the-art in natural image denoising; however, it is targeted primarily toward Gaussian noise. In our preliminary evaluation, BM3D did not perform well on many of the variety of noise types.

While BM3D is a well-engineered algorithm, Burger et al. [9] showed that it is possible to achieve state-of-the-art denoising performance with a plain multi-layer perceptron (MLP) that maps noisy patches onto noise-free ones, once the capacity of the MLP, the patch size, and the training set are large enough. Therefore, neural networks indeed have a great potential for image denoising.

Vincent et al. [29] introduced the stacked denoising autoencoders as a way of providing a good initial representation of the data in deep networks for classification tasks. Our proposed AMC-SSDA builds upon this work by using the denoising autoencoder's internal representation to determine the optimal column weighting for robust denoising.

Cireşan et al. [10] presented a multi-column approach for image classification, averaging the output of several deep neural networks (or *columns*) trained on inputs preprocessed in different ways. However, based on our experiments, this approach (i.e., simply averaging the output of each column) is not robust in denoising since each column has been trained on a different type of noise. To address this problem, we propose an adaptive weighting scheme that can handle a variety of noise types.

Jain et al. [18] used deep convolutional neural networks for image denoising. Rather than using a convolutional approach, our proposed method applies multiple sparse autoencoder networks in combination to the denoising task. Tang et al. [28] applied deep learning techniques (e.g., extensions of the deep belief network with local receptive fields) to denoising and classifying MNIST digits. In comparison, we achieve favorable classification performance on corrupted MNIST digits.

## 3   Algorithm

In this section, we first describe the SSDA [31]. Then we will present the AMC-SSDA and describe the process of finding optimal column weights and predicting column weights for test images.

### 3.1   Stacked sparse denoising autoencoders

A denoising autoencoder (DA) [29] is typically used as a way to pre-train layers in a deep neural network, avoiding the difficulty in training such a network as a whole from scratch by performing greedy layer-wise training (e.g., [4, 5, 14]). As Xie et al. [31] showed, a denoising autoencoder is

also a natural fit for performing denoising tasks, due to its behavior of taking a noisy signal as input and reconstructing the original, clean signal as output.

Commonly, a series of DAs are connected to form a stacked denoising autoencoder (SDA)—a deep network formed by feeding the hidden layer's activations of one DA into the input of the next DA. Typically, SDAs are pre-trained in an unsupervised fashion where each DA layer is trained by generating new noise [29]. We follow Xie et al.'s method of SDA training by calculating the first layer activations for both the clean input and noisy input to use as training data for the second layer. As they showed, this modification to the training process allows the SDA to better learn the features for denoising the original corrupting noise.

More formally, let $\mathbf{y} \in \mathbb{R}^D$ be an instance of uncorrupted data and $\mathbf{x} \in \mathbb{R}^D$ be the corrupted version of $\mathbf{y}$. We can define the feedforward functions of the DA with $K$ hidden units as follows:

$$\mathbf{h}(\mathbf{x}) = f(\mathbf{Wx} + \mathbf{b}) \tag{1}$$

$$\hat{\mathbf{y}}(\mathbf{x}) = g(\mathbf{W}'\mathbf{h} + \mathbf{b}') \tag{2}$$

where $f()$ and $g()$ are respectively encoding and decoding functions (for which sigmoid function $\sigma(s) = \frac{1}{1+\exp(-s)}$ is often used),[1] $\mathbf{W} \in \mathbb{R}^{K \times D}$ and $\mathbf{b} \in \mathbb{R}^K$ are encoding weights and biases, and $\mathbf{W}' \in \mathbb{R}^{D \times K}$ and $\mathbf{b}' \in \mathbb{R}^D$ are the decoding weights and biases. $\mathbf{h}(\mathbf{x}) \in \mathbb{R}^K$ is the hidden layer's activation, and $\hat{\mathbf{y}}(\mathbf{x}) \in \mathbb{R}^D$ is the reconstruction of the input (i.e., the DA's output). Given training data $\mathcal{D} = \{(\mathbf{x}_1, \mathbf{y}_1), ..., (\mathbf{x}_N, \mathbf{y}_N)\}$ with $N$ training examples, the DA is trained by back-propagation to minimize the sparsity regularized reconstruction loss given by

$$\mathcal{L}_{\text{DA}}(\mathcal{D}; \boldsymbol{\Theta}) = \frac{1}{N} \sum_{i=1}^{N} \|\mathbf{y}_i - \hat{\mathbf{y}}(\mathbf{x}_i)\|_2^2 + \beta \sum_{j=1}^{K} \text{KL}(\rho \| \hat{\rho}_j) + \frac{\lambda}{2}(\|\mathbf{W}\|_{\text{F}}^2 + \|\mathbf{W}'\|_{\text{F}}^2) \tag{3}$$

where $\boldsymbol{\Theta} = \{\mathbf{W}, \mathbf{b}, \mathbf{W}', \mathbf{b}'\}$ are the parameters of the model, and the sparsity-inducing term $\text{KL}(\rho \| \hat{\rho}_j)$ is the Kullback-Leibler divergence between $\rho$ (target activation) and $\hat{\rho}_j$ (empirical average activation of the $j$-th hidden unit) [20, 13]:

$$\text{KL}(\hat{\rho}_j \| \rho) = \rho \log \frac{\rho}{\hat{\rho}_j} + (1 - \rho) \log \frac{(1 - \rho)}{1 - \hat{\rho}_j} \quad \text{where} \quad \hat{\rho}_j = \frac{1}{N} \sum_{i=1}^{N} h_j(\mathbf{x}_i) \tag{4}$$

and $\lambda$, $\beta$, and $\rho$ are scalar-valued hyperparameters determined by cross validation.

In this work, two DAs are stacked as shown in Figure 1a, where the activation of the first DA's hidden layer provides the input to the second DA, which in turn provides the input to the output layer of the first DA. This entire network—the SSDA—is trained again by back-propagation in a fine tuning stage, minimizing the loss given by

$$L_{\text{SSDA}}(\mathcal{D}; \boldsymbol{\Theta}) = \frac{1}{N} \sum_{i=1}^{N} \|\mathbf{y}_i - \hat{\mathbf{y}}(\mathbf{x}_i)\|_2^2 + \frac{\lambda}{2} \sum_{l=1}^{2L} \|\mathbf{W}^{(l)}\|_{\text{F}}^2 \tag{5}$$

where $L$ is the number of stacked DAs (we used $L = 2$ in our experiments), and $\mathbf{W}^{(l)}$ denotes weights for the $l$-th layer in the stacked deep network.[2] The sparsity-inducing term is not needed for this step because the sparsity was already incorporated in the pre-trained DAs. Our experiments show that there is not a significant change in performance when sparsity is included.

### 3.2 Adaptive Multi-Column SSDA

The adaptive multi-column SSDA is the linear combination of several SSDAs, or *columns*, each trained on a single type of noise using optimized weights determined by the features of each given input image. Taking advantage of the SSDA's capability of feature learning, we use the features generated by the activation of the SSDA's hidden layers as inputs to a neural network-based regression component, referred to here as the *weight prediction module*. As shown in Figure 1b, this module then uses these features to compute the optimal weights used to linearly combine the column outputs into a weighted average.

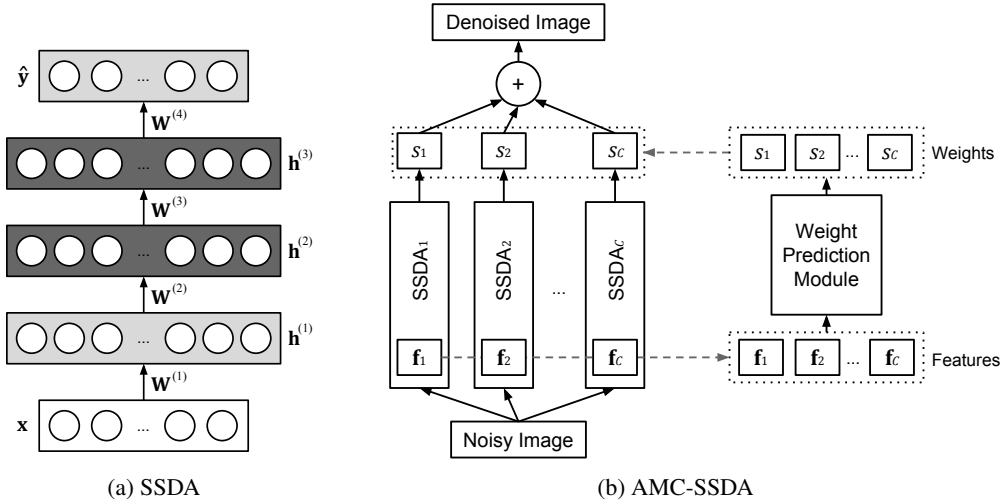

(a) SSDA

(b) AMC-SSDA

Figure 1: Illustration of the AMC-SSDA. We concatenate the activations of the first-layer hidden units of the SSDA in each column (i.e., $\mathbf{f}_c$ denotes the concatenated hidden unit vectors $\mathbf{h}^{(1)}(\mathbf{x})$ and $\mathbf{h}^{(2)}(\mathbf{x})$ of the SSDA corresponding to $c$-th column) as input features to the weight prediction module for determining the optimal weight for each column of the AMC-SSDA. See text for details.

### 3.2.1 Training the AMC-SSDA

The AMC-SSDA has three training phases: training the SSDAs, determining optimal weights for a set of training images, and then training the weight prediction module. The SSDAs are trained as discussed in Section 3.1, with each SSDA provided a noisy training set, corrupted by a single noise type along with the original versions of those images as a target set. Each SSDA column $c$ then produces an output $\hat{\mathbf{y}}_c \in \mathbb{R}^D$ for an input $\mathbf{x} \in \mathbb{R}^D$, which is the noisy version of original image $\mathbf{y}$. (We omit index $i$ to remove clutter.)

### 3.2.2 Finding optimal column weights via quadratic program

Once the SSDAs are trained, we construct a new training set that pairs features extracted from the hidden layers of the SSDAs with optimal column weights. Specifically, for each image, a vector $\phi = [\mathbf{f}_1; ...; \mathbf{f}_C]$ is built from the features extracted from the hidden layers of each SSDA, where $C$ is the number of columns. That is, for SSDA column $c$, the activations of hidden layers $\mathbf{h}^{(1)}$ and $\mathbf{h}^{(2)}$ (as shown in Figure 1a) are concatenated into a vector $\mathbf{f}_c$, and then $\mathbf{f}_1, \mathbf{f}_2, \ldots, \mathbf{f}_C$ are concatenated to form the whole feature vector $\phi$.

Additionally, the output of each column for each image is collected into a matrix $\hat{\mathbf{Y}} = [\mathbf{y}_1, ..., \mathbf{y}_C] \in \mathbb{R}^{D \times C}$, with each column being the output of one of the SSDA columns, $\hat{\mathbf{y}}_c$. To determine the ideal linear weighting of the SSDA columns for that given image, we perform the following non-linear minimization (quadratic program) as follows:[3]

$$\text{minimize}_{\{s_c\}} \quad \frac{1}{2}\|\hat{\mathbf{Y}}\mathbf{s} - \mathbf{y}\|^2 \tag{6}$$

$$\text{subject to} \quad 0 \le s_c \le 1, \forall c \tag{7}$$

$$1 - \delta \le \sum_{c=1}^{C} s_c \le 1 + \delta \tag{8}$$

Here $\mathbf{s} \in \mathbb{R}^C$ is the vector of weights $s_c$ corresponding to each SSDA column $c$. Constraining the weights between 0 and 1 was shown to allow for better weight predictions by reducing overfitting. The constraint in Eq. (8) helps to avoid degenerate cases where weights for very bright or dark spots

| Noise Type | Parameter | Parameter value |
|---|---|---|
| Gaussian | $\sigma^2$ | 0.02, 0.06, 0.10, 0.14, 0.18, 0.22, 0.26 |
| Speckle | $\rho$ | 0.05, 0.10, 0.15, 0.20, 0.25, 0.30, 0.35 |
| Salt & Pepper | $\rho$ | 0.05, 0.10, 0.15, 0.20, 0.25, 0.30, 0.35 |

Table 1: SSDA training noises in the 21-column AMC-SSDA. $\rho$ is the noise density.

would otherwise be very high or low. Although making the weights sum exactly to one is more intuitive, we found that the performance slightly improved when given some flexibility, as shown in Eq. (8). For our experiments, $\delta = 0.05$ is used.

### 3.2.3 Learning to predict optimal column weights via RBF networks

The final training phase is to train the weight prediction module. A radial basis function (RBF) network is trained to take the feature vector $\phi$ as input and produce a weight vector $\mathbf{s}$, using the optimal weight training set described in Section 3.2.2. An RBF network was chosen for our experiments because of its known performance in function approximation [24]. However, other function approximation techniques could be used in this step.

### 3.2.4 Denoising with the AMC-SSDA

Once training has been completed, the AMC-SSDA is ready for use. A noisy image $\mathbf{x}$ is supplied as input to each of the columns, which together produce the output matrix $\hat{\mathbf{Y}}$, each column of which is the output of a particular column of the AMC-SSDA. The feature vector $\phi$ is created from the activation of the hidden layers of each SSDA (as described in Section 3.2.2) and fed into the weight prediction module (as described in Section 3.2.3), which then computes the predicted column weights, $\mathbf{s}^*$. The final denoised image $\hat{\mathbf{y}}$ is produced by linearly combining the columns using these weights: $\hat{\mathbf{y}} = \hat{\mathbf{Y}}\mathbf{s}^*$.[4]

## 4 Experiments

We performed a number of denoising tasks by corrupting and denoising images of computed tomography (CT) scans of the head from the Cancer Imaging Archive [17] (Section 4.1). Quantitative evaluation of denoising results was performed using peak signal-to-noise ratio (PSNR), a standard method used for evaluating denoising performance. PSNR is defined as $\mathrm{PSNR} = 10\log_{10}(p_{\max}^2/\sigma_e^2)$, where $p_{\max}$ is the maximum possible pixel value and $\sigma_e^2$ is the mean-square error between the noisy and original images. We also tested the AMC-SSDA as pre-processing step in an image classification task by corrupting MNIST database of handwritten digits [19] with various types of noise and then denoising and classifying the digits with a classifier trained on the original images (Section 4.2).

Our code is available at: http://sites.google.com/site/nips2013amcssda/.

### 4.1 Image denoising

To evaluate general denoising performance, images of CT scans of the head were corrupted with seven variations of Gaussian, salt-and-pepper, and speckle noise, resulting in the 21 noise types shown in Table 1. Twenty-one individual SSDAs were trained on randomly selected 8-by-8 pixel patches from the corrupted images; each SSDA was trained on a single type of noise. These twenty-one SSDAs were used as columns to create an AMC-SSDA.[5] The testing noise is given in Table 2. The noise was produced using Matlab's `imnoise` function with the exception of uniform noise, which was produced with our own implementation. For Poisson noise, the image is divided by $\lambda$ prior to applying the noise; the result is then multiplied by $\lambda$.

To train the weight predictor for the AMC-SSDA, a set of images disjoint from the training set of the individual SSDAs were used. The training images for the AMC-SSDA were corrupted with the same noise types used to train its columns. The AMC-SSDA was tested on another set of images

| Noise Type | 1 | 2 | 3 | 4 |
|---|---|---|---|---|
| Gaussian | $\sigma^2 = 0.01$ | $\sigma^2 = 0.07$ | $\sigma^2 = 0.1$ | $\sigma^2 = 0.25$ |
| Speckle | $\rho = 0.1$ | $\rho = 0.15$ | $\rho = 0.3$ | $\rho = 0.4$ |
| Salt & Pepper | $\rho = 0.1$ | $\rho = 0.15$ | $\rho = 0.3$ | $\rho = 0.4$ |
| Poisson | $\log(\lambda) = 24.4$ | $\log(\lambda) = 25.3$ | $\log(\lambda) = 26.0$ | $\log(\lambda) = 26.4$ |
| Uniform [-0.5, 0.5] | 30% | 50% | 70% | 100% |

Table 2: Parameters of noise types used for testing. The Poisson and uniform noise types are not seen in the training set. The percentage for uniform noise denotes how many pixels are affected. $\rho$ is the noise density.

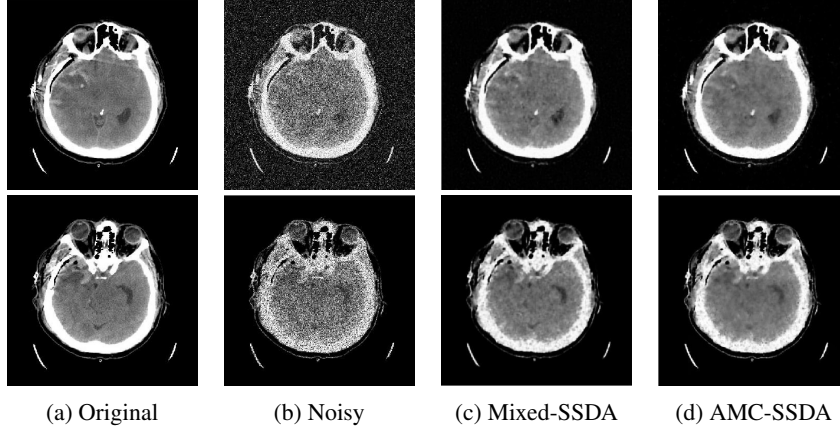

     (a) Original       (b) Noisy       (c) Mixed-SSDA       (d) AMC-SSDA

Figure 2: Visualization of the denoising performance of the Mixed-SSDA and AMC-SSDA. Top: Gaussian noise. Bottom: speckle noise.

disjoint from both the individual SSDA and AMC-SSDA training sets. The AMC-SSDA was trained on 128-by-128 pixel patches. When testing, 64-by-64 pixel patches are denoised with a stride of 48. During testing, we found that smaller strides yielded a very small increase in PSNR; however, having a small stride was not feasible due to memory constraints. Since our SSDAs denoise 8-by-8 patches, features for, say, a 64-by-64 patch are the average of the features extracted for each 8-by-8 patch in the 64-by-64 patch. We find that this allows for more consistent and predictable weights. The AMC-SSDA is first tested on noise types that have been seen (i.e., noise types that were in the training set) but have different statistics. It is then tested on noise not seen in the training examples, referred to as "unseen" noise.

To compare with the experiments of Xie et al. [31], one SSDA was trained on only the Gaussian noise types, one on only salt & pepper, one on only speckle, and one on all the noise types from Table 1. We refer to these as *gaussian SSDA*, *s&p SSDA*, *speckle SSDA*, and *mixed SSDA*, respectively. These SSDAs were then tested on the same types of noise that the AMC-SSDA was tested on. The results for both seen and unseen noise can be found in Tables 3 and 4. On average, for all cases, the AMC-SSDA produced superior PSNR values when compared to these SSDAs. Some example results are shown in Figure 2. In addition, we test the case where all the weights are equal and sum to one. We call this the MC-SSDA; note that there is no adaptive element to it. We found that AMC-SSDA also outperformed MC-SSDA.

**Statistical significance** We statistically evaluated the difference between our AMC-SSDA and the mixed SSDA (the best performing SSDA baseline) for the results shown in Table 3, using the one-tailed paired t-test. The AMC-SSDA was significantly better than the mixed-SSDA, with a p-value of $3.3 \times 10^{-5}$ for the null hypothesis. We also found that even for a smaller number of columns (such as 9 columns), the AMC-SSDA still was superior to the mixed-SSDA with statistical significance. In this paper, we report results from the 21-column AMC-SSDA.

We also performed additional control experiments in which we gave the SSDA an unfair advantage. Specifically, each test image corrupted with seen noise was denoised with an SSDA that had been trained on the exact type of noise and statistics that the test image has been corrupted with; we call this the "informed-SSDA." We saw that the AMC-SSDA performed slightly better on the Gaussian

| Noise Type | Noisy Image | Gaussian SSDA | S&P SSDA | Speckle SSDA | Mixed SSDA | MC-SSDA | AMC-SSDA |
|---|---|---|---|---|---|---|---|
| G 1 | 22.10 | 26.64 | 26.69 | 26.84 | 27.15 | 27.37 | **29.60** |
| G 2 | 13.92 | 25.83 | 23.07 | 19.76 | 25.52 | 23.34 | **26.85** |
| G 3 | 12.52 | 25.50 | 22.17 | 18.35 | 25.09 | 22.00 | **26.10** |
| G 4 | 9.30 | 23.11 | 20.17 | 14.88 | 22.72 | 17.97 | **23.66** |
| SP 1 | 13.50 | 25.86 | 26.26 | 22.27 | 26.32 | 25.84 | **27.72** |
| SP 2 | 11.76 | 25.40 | 25.77 | 20.07 | 25.77 | 24.54 | **26.77** |
| SP 3 | 8.75 | 23.95 | 23.96 | 15.88 | 24.32 | 20.42 | **24.65** |
| SP 4 | 7.50 | 22.46 | 22.20 | 13.86 | 22.95 | 17.76 | **23.01** |
| S 1 | 19.93 | 26.41 | 26.37 | 28.22 | 26.97 | 27.43 | **28.59** |
| S 2 | 18.22 | 25.92 | 25.80 | **27.75** | 26.44 | 26.71 | 27.68 |
| S 3 | 15.35 | 23.54 | 23.36 | **25.79** | 24.42 | 23.91 | 25.72 |
| S 4 | 14.24 | 21.80 | 21.69 | **24.41** | 22.93 | 22.20 | 24.35 |
| Avg | 13.92 | 24.70 | 23.96 | 21.51 | 25.05 | 23.29 | **26.23** |

(a) PSNRs for previously seen noise, best values in bold.

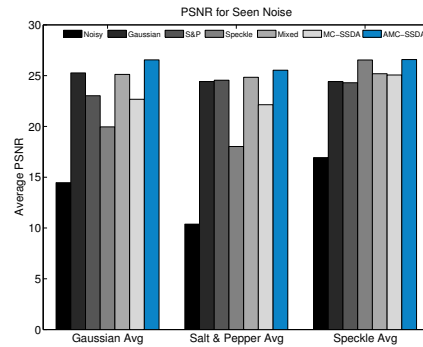

(b) Average PNSRs for specific noise types

Figure 3: Average PSNR values for denoised images of various previously seen noise types (G: Gaussian, S: Speckle, SP: Salt & Pepper).

| Noise Type | Noisy Image | Gaussian SSDA | S&P SSDA | Speckle SSDA | Mixed SSDA | MC-SSDA | AMC-SSDA |
|---|---|---|---|---|---|---|---|
| P 1 | 19.90 | 26.27 | 26.48 | 27.99 | 26.80 | 27.35 | **28.83** |
| P 2 | 16.90 | 25.77 | 25.92 | 26.94 | 26.01 | 26.78 | **27.64** |
| P 3 | 13.89 | 24.61 | 24.54 | 24.65 | 24.43 | 25.11 | **25.50** |
| P 4 | 12.11 | 23.36 | 23.07 | 22.64 | 23.01 | 23.28 | **23.43** |
| U 1 | 17.20 | 23.40 | 23.68 | **25.05** | 23.74 | 24.71 | 24.50 |
| U 2 | 16.04 | 26.21 | 25.86 | 23.21 | 26.28 | 26.13 | **28.06** |
| U 3 | 12.98 | 23.24 | 21.36 | 17.83 | 22.89 | 21.07 | **23.70** |
| U 4 | 8.78 | 16.54 | 15.45 | 12.01 | 16.04 | 14.11 | **16.78** |
| Avg | 14.72 | 23.67 | 23.29 | 22.54 | 23.65 | 23.57 | **24.80** |

(a) PSNR for unseen noise, best values in bold.

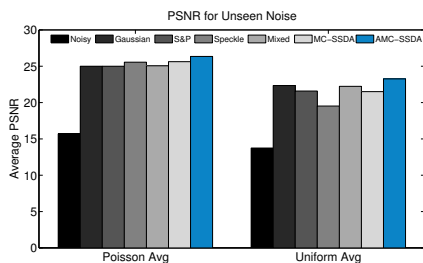

(b) Average results for noise types.

Figure 4: Average PSNR values for denoised images of various previously unseen noise types (P: Poisson noise; U: Uniform noise).

and salt & pepper noise and slightly worse on speckle noise. Overall, the informed-SSDA had, on average, a PSNR that was only 0.076dB better than the AMC-SSDA. The p-value obtained was 0.4708, indicating little difference between the two methods. This suggests that the AMC-SSDA can perform as well as using an "ideally" trained network for specific noise type (i.e., training and testing an SSDA for the same specific noise type). This is achieved through its adaptive functionality.

## 4.2 Digit recognition from denoised images

Since the results of denoising images from a visual standpoint can be more qualitative than quantitative, we have tested using denoising as a preprocessing step done before a classification task. Specifically, we used the MNIST database of handwritten digits [19] as benchmark to evaluate the efficacy of our denoising procedures.

First, we trained a deep neural network digit classifier from the MNIST training digits, following [15]. The digit classifier achieved a baseline error rate of 1.09% when tested on the uncorrupted MNIST test set.

The MNIST digits are corrupted with Gaussian, salt & pepper, speckle, block, and border noise. Examples of this are shown in Figure 5. The block and border noises are similar to that of Tang

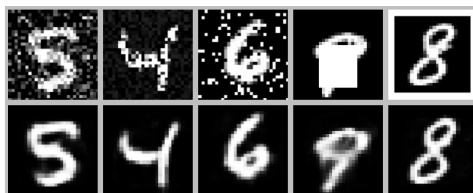

Figure 5: Example MNIST digits. Noisy images are shown on top and the corresponding denoised images by the AMC-SSDA are shown below. Noise types from left: Gaussian, speckle, salt & pepper, block, border.

et al. [28]. An SSDA was trained on each type of noise. An AMC-SSDA was also trained using these types of noise. The goal of this experiment is to show that the potential cumbersome and time-consuming process of determining the type of noise that an image is corrupted with at test time is not needed to achieve good classification results.

As the results show in Table 3, the denoising performance was strongly correlated to the type of noise upon which the denoiser was trained. The bold-faced values show the best performing denoiser for a given noise type. Since a classification difference of 0.1% or larger is considered statistically significant [5], we bold all values within 0.1% of the best error rate. The AMC-SSDA either outperforms, or comes close to (within 0.06%), the SSDA that was trained with the same type of noise as in the test data. In terms of average error across all types of noises, the AMC-SSDA is significantly better than any single denoising algorithms we compared. The results suggest that the AMC-SSDA consistently achieves strong classification performance without having to determine the type of noise during test time.

These results are also comparable to the results of Tang et al. [28]. We show that we get better classification accuracy for the block and border noise types. In addition, we note that Tang et al. uses a 7-by-7 local receptive field, while ours uses 28-by-28 patches. As suggested by Tang et al., we expect that using a local field in our architecture could further improve our results.

| Method / Noise Type | Clean | Gaussian | S & P | Speckle | Block | Border | Average |
|---|---|---|---|---|---|---|---|
| No denoising | **1.09**% | 29.17% | 18.63% | 8.11% | 25.72% | 90.05% | 28.80% |
| Gaussian SSDA | 2.13% | **1.52**% | 2.44% | 5.10% | 20.03% | 8.69% | 6.65% |
| Salt & Pepper SSDA | 1.94% | 1.71% | 2.38% | 4.78% | 19.71% | 2.16% | 5.45% |
| Speckle SSDA | 1.58% | 5.86% | 6.80% | **2.03**% | 19.95% | 7.36% | 7.26% |
| Block SSDA | 1.67% | 5.92% | 15.29% | 7.64% | **5.15**% | 1.81% | 6.25% |
| Border SSDA | 8.42% | 19.87% | 19.45% | 13.89% | 31.38% | **1.12**% | 15.69% |
| AMC-SSDA | 1.50% | **1.47**% | **2.22**% | **2.09**% | **5.18**% | **1.15**% | **2.27**% |
| Tang et al. [28]* | 1.24% | - | - | - | 19.09% | 1.29% | - |

Table 3: MNIST test classification error of denoised images. Rows denote the performance of different denoising methods, including: "no denoising," SSDA trained on a specific noise type, and AMC-SSDA. Columns represent images corrupted with the given noise type. Percentage values are classification error rates for a set of test images corrupted with the given noise type and denoised prior to classification. Bold-faced values represent the best performance for images corrupted by a given noise type. *Note: we compare the numbers reported from Tang et al. [28] ("7x7+denoised").

## 5 Conclusion

In this paper, we proposed the adaptive multi-column SSDA, a novel technique of combining multiple SSDAs by predicting optimal column weights adaptively. We have demonstrated that AMC-SSDA can robustly denoise images corrupted by multiple different types of noise without knowledge of the noise type at testing time. It has also been shown to perform well on types of noise that were not in the training set. Overall, the AMC-SSDA has significantly outperformed the SSDA in denoising. The good classification results of denoised MNIST digits also support the hypothesis that the AMC-SSDA eliminates the need to know about the type of noise during test time.

**Acknowledgments**

This work was funded in part by Google Faculty Research Award, ONR N00014-13-1-0762, and NSF IIS 1247414. F. Agostinelli was supported by GEM Fellowship, and M. Anderson was supported in part by NSF IGERT Open Data Fellowship (#0903629). We also thank Roni Mittelman, Yong Peng, Scott Reed, and Yuting Zhang for their helpful comments.

## Footnotes

[1]In particular, the sigmoid function is often used for decoding the input data when their values are bounded between 0 and 1. For general cases, other types of functions (such as $\tanh$, rectified linear, or linear functions) can be used.

[2]After pre-training, we initialized $\mathbf{W}^{(1)}$ and $\mathbf{W}^{(4)}$ from the encoding and decoding weights of the first-layer DA, and $\mathbf{W}^{(2)}$ and $\mathbf{W}^{(3)}$ from the encoding and decoding weights of the second-layer DA, respectively.

[3]In addition to the L2 error shown in Equation (6), we also tested minimizing the L1 distance as the error function, which is a standard method in the related field of image registration [3]. The version using the L1 error performed slightly better in our noisy digit classification task, suggesting that the loss function might need to be tuned to the task and images at hand.

[4]We have tried alternatives to this approach. Some of these involved using a single unified network to combine the columns, such as joint training. In our preliminary experiments, these approaches did not yield significant improvements.

[5]We also evaluated AMC-SSDAs with smaller number of columns. In general, we achieved better performance with more columns. We discuss its statistical significance later in this section.

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
