[Reviews · NeurIPS 2013]

Submitted by Assigned_Reviewer_5

This paper uses deep neural networks to denoise images. The network uses several independent columns trained on different types of noise. Furthermore, a weight prediction module is trained to decide how to merge these columns to predict the clean image from the noisy image. The idea is to have only one network for all types of noises and noise strengths.

Noise is not always Gaussian, and it may sometimes be beneficial to have a single method that can adapt to different noise types. Engineered methods like BM3D are hard to modify to arbitrary noise types, learned methods like Burger et al. have to be retrained (or also the missing citation Jancsary et al., "Loss-Speci fic Training of Non-Parametric Image Restoration Models").
This method solves it by training different models on different noises and then combining the results. The method is shown to be successful in denoising different types of noises after a one-time training. Denoising also works on "unseen" noise that wasn't trained on in any of its submodels.
However, it is inferior to just training on a single noise type for that specific noise type: SSDA outperforms MC-SSDA on the CT images on Gaussian noise type 2 and 3, which have similar PSNR to the training PSNR of SSDA. The same holds true for the MNIST comparison. Furthermore, the SSDA is also able to denoise unseen noise types to some extent, in one case even better. [The authors were able to address this issue during the rebuttal by training a new model with additional columns.]
Also, it's not compared to standard denoising methods on natural images, which I would expect. On digital photos the noise usually is a mixture of Gaussian and Poisson noise, which the method should be able to handle.
I wonder what the runtime of the method is (training/testing)? Also, how does the performance change if the number of autoencoders is varied?

The paper is well structured and easy to understand. Very beneficial is that the code would be published after publication.
Summary: This paper learns a single neural network to denoise several types of noises. This works, however it's a trade-off compared to using an algorithm for one specific noise type.

Submitted by Assigned_Reviewer_6

This paper explores combining multiple towers of stacked sparse denoising autoencoders. Simple averaging of towers like in Ciresan does for classification cannot work for denoising so authors instead predict the weights used to linearly combine the towers. The weight predictor is a RBF network with regression given the tower features.

Experiments with both qualitative and quantitative results are convincing (classification and denoising) and MC-SSDA does provide a good improvement over single SSDA.

Quality: good, multiple experiments, different tasks.
Clarity: good.
Originality: reusing non-novel pieces and multiple towers have been used for classification, but I don't know that it has for denoising.
Significance: might be of significance for medical imaging community.
Summary: Good paper, no new ideas but good results.

Submitted by Assigned_Reviewer_7

This paper proposes the use of multiple columns (networks), with proper weighting, for image denoising. The multiple columns allow for the ability to denoise many different types of noise without knowing which noise type at test time.

An RBF network is learned to predict the weighting of the contribution of different columns, which is similar to mixture of experts.

Line 88-89: The citation of Hinton et al. transforming autoencoders is not appropriate here. Transforming autoencoders is about learning transformations and have nothing to do with denoising.

A criticism of this method is that you are training with known noise processes. A lot of the work in denoising literature is learning models without having the ability to generate clean-corrupt pairs of training images. Another is that the network is c times slower, where c is the number of the columns.

The experimental results demonstrate good performance on MNIST images and medical image datasets. An additional recommendation is to use real noise instead of synthetic noise added by 'imnoise'.
Summary: This paper demonstrates superior performance in denoising by using multiple columns of neural net to handle multiple types of noise. Experimental results show improvement over the single column variants.
Author Feedback

Author rebuttal: We thank all reviewers for helpful comments.

1. Novelty of our method:
As Reviewer 6 pointed out, the conventional way of combining multi-column deep networks by simple averaging does not work well when the distribution of training data does not match the distribution of test data. The key novelty of our method is to address this limitation by (1) computing optimal column weights via solving a quadratic program and (2) training a separate network to predict the optimal weights. Our method is generally applicable to not only denoising, but also many other problems.


2. Regarding comparison to SSDA:
To address the concern that our method was a "trade-off compared to using an algorithm for one specific type of noise," we trained a 21 column AMC-SSDA. In our experiments, this single AMC-SSDA outperformed (on average) other baseline SSDAs trained from any specific type of noise (e.g., Gaussian, salt & pepper, or speckle) or the mixture of all these noise types.

In additional control experiment, for each "seen" noise type that was tested on in the paper, we trained an SSDA with that exact noise type, including the exact statistics of the noise; let's call this the "informed-SSDA". Please note that this setting provides an unfair advantage to the informed-SSDA since we do not provide information about the noise type or statistics in testing time to any other methods evaluated in the paper.

Even in this case, the AMC-SSDA still performed comparably to the informed-SSDA. Specifically, the AMC-SSDA performed slightly better on the Gaussian noise or Salt & Pepper noise, and slightly worse for Speckle noise. On average, the informed-SSDA had a PSNR that was only slightly better than the AMC-SSDA. Using the two-tailed paired t-test, the p-value we obtained was 0.47, showing that there is not a statistically significant difference between these two methods. This suggests that the AMC-SSDA can perform as well as using an "ideally" trained network for specific noise type (i.e., training and testing an SSDA for the same specific noise type).

In addition, our single AMC-SSDA achieved classification error rates better than (up to 0.22%) or comparable to (within 0.06% difference) the corresponding informed-SSDAs.


3. Performance with varying number of columns:
In our experiments, we found that the increased number of column leads to better performance. For example, the AMC-SSDA with 21 columns had, on average, a PSNR that was significantly higher than the AMC-SSDA with 9 columns (i.e., 0.45dB higher on average, with p-value of 0.001).